# A Machine Learning-Based Multiple Imputation Method for the Health and Aging Brain Study–Health Disparities

**Fan Zhang** [1,2,*], **Melissa Petersen** [1,2], **Leigh Johnson** [1,3], **James Hall** [1,2], **Raymond F. Palmer** [4], **Sid E. O'Bryant** [1,2] **and on behalf of the Health and Aging Brain Study (HABS–HD) Study Team** [†]

1 Institute for Translational Research, University of North Texas Health Science Center, Fort Worth, TX 76107, USA; melissa.petersen@unthsc.edu (M.P.); leigh.johnson@unthsc.edu (L.J.); james.hall@unthsc.edu (J.H.); sid.obryant@unthsc.edu (S.E.O.)
2 Department of Family Medicine, University of North Texas Health Science Center, Fort Worth, TX 76107, USA
3 Department of Pharmacology and Neuroscience, University of North Texas Health Science Center, Fort Worth, TX 76107, USA
4 Department of Family and Community Medicine, University of Texas Health Science Center, San Antonio, TX 78229, USA; palmerr@uthscsa.edu
* Correspondence: fan.zhang@unthsc.edu; Tel.: +1-817-735-2947
† HABS–HD MPIs: Sid E O'Bryant, Kristine Yaffe, Arthur Toga, Robert Rissman, & Leigh Johnson; and the HABS–HD Investigators: Meredith Braskie, Kevin King, James R Hall, Melissa Petersen, Raymond Palmer, Robert Barber, Yonggang Shi, Fan Zhang, Rajesh Nandy, Roderick McColl, David Mason, Bradley Christian, Nicole Philips, Stephanie Large, Joe Lee, Badri Vardarajan, Monica Rivera Mindt, Amrita Cheema, Lisa Barnes, Mark Mapstone, Annie Cohen, Amy Kind, Ozioma Okonkwo, Raul Vintimilla, Zhengyang Zhou, Michael Donohue, Rema Raman, Matthew Borzage, Michelle Mielke, Beau Ances, Ganesh Babulal, Jorge Llibre-Guerra, Carl Hill and Rocky Vig.

**Abstract:** The Health and Aging Brain Study–Health Disparities (HABS–HD) project seeks to understand the biological, social, and environmental factors that impact brain aging among diverse communities. A common issue for HABS–HD is missing data. It is impossible to achieve accurate machine learning (ML) if data contain missing values. Therefore, developing a new imputation methodology has become an urgent task for HABS–HD. The three missing data assumptions, (1) missing completely at random (MCAR), (2) missing at random (MAR), and (3) missing not at random (MNAR), necessitate distinct imputation approaches for each mechanism of missingness. Several popular imputation methods, including listwise deletion, min, mean, predictive mean matching (PMM), classification and regression trees (CART), and missForest, may result in biased outcomes and reduced statistical power when applied to downstream analyses such as testing hypotheses related to clinical variables or utilizing machine learning to predict AD or MCI. Moreover, these commonly used imputation techniques can produce unreliable estimates of missing values if they do not account for the missingness mechanisms or if there is an inconsistency between the imputation method and the missing data mechanism in HABS–HD. Therefore, we proposed a three-step workflow to handle missing data in HABS–HD: (1) missing data evaluation, (2) imputation, and (3) imputation evaluation. First, we explored the missingness in HABS–HD. Then, we developed a machine learning-based multiple imputation method (MLMI) for imputing missing values. We built four ML-based imputation models (support vector machine (SVM), random forest (RF), extreme gradient boosting (XGB), and lasso and elastic-net regularized generalized linear model (GLMNET)) and adapted the four ML-based models to multiple imputations using the simple averaging method. Lastly, we evaluated and compared MLMI with other common methods. Our results showed that the three-step workflow worked well for handling missing values in HABS–HD and the ML-based multiple imputation method outperformed other common methods in terms of prediction performance and change in distribution and correlation. The choice of missing handling methodology has a significant impact on the accompanying statistical analyses of HABS–HD. The conceptual three-step workflow and the ML-based multiple imputation method perform well for our Alzheimer's disease models. They can also be applied to other disease data analyses.

**Keywords:** Alzheimer's disease; blood biomarkers; machine learning; multiple imputation; missing data

## 1. Introduction

In the US, Alzheimer's disease (AD) is the most common type of dementia, accounting for 60–80% of dementia cases. Currently, more than 6 million Americans have Alzheimer's, and by 2050, this number is estimated to rise to 13 million [1].

The Health and Aging Brain Study–Health Disparities (HABS–HD) project seeks to understand the biological, social, and environmental factors that impact brain aging among diverse communities [2]. HABS–HD, like many other NIH-funded data-sharing projects, has important data assets for various uses, including social, environmental, and behavioral data and multiple data flow pathways. A common methodological issue for data collection in such a large survey-based, biological, behavioral, or environmental epidemiology study are missing data.

There were several ways that missingness could occur during biomarker collection in HABS–HD. Biomarkers were typically measured using electrochemiluminescence (ECL) and the Quanterix Simoa HD-1 platform [2]. Although both the platforms exhibited excellent performance (CVs $\leq$ 10% for ECL and CVs $\leq$ 5% for Quanterix Simoa HD-1), these protein profiling instruments often introduced missing values into the data. Firstly, the signal of a protein might be lower than the instrument's detection limit. Secondly, proteomic profiling of a tissue or cell line often resulted in the absence of a substantial proportion of proteins. Thirdly, various factors related to the measurement process, such as the instrument used, batch effects, or variations in bioinformatics processing pipelines, might result in missing values. Finally, missing values could occur randomly in one of the three areas listed above.

The missing data in HABS–HD are important for two reasons. First, the problem is difficult to identify. Each variable may only have a small number of missing responses, but in combination, the missing data could be numerous. Only thorough analysis of missing data can determine whether missing data are problematic. However, this analysis can be time-consuming and error-prone. Second, it can cause serious problems. For example, most statistical procedures automatically eliminate cases with missing data. This may cause data loss for analysis and make results misleading because the analyzed cases are not a random sample of all cases. Therefore, understanding and properly handling missing data in HABS–HD is crucial for obtaining accurate and meaningful results from machine learning (ML) [3,4].

When it comes to missing data, there are three mechanisms to consider: missing completely at random (MCAR), missing at random (MAR), and missing not at random (MNAR) [5–7]. Different mechanisms of missingness require different methods to handle because different approaches make different assumptions [3,4,8]. Some commonly used methods, for example, listwise deletion, min/mean substitution, and predictive mean matching (PMM) [9,10], might introduce bias, lead to incorrect inference and interpretations, and reduce statistical power in downstream analyses such as hypothesis testing of associations with clinical variables or machine learning predicting AD or MCI. First, listwise deletion can lead to biased results if the missing values are not missing completely at random (MCAR) [3]. This is because the missing values may be associated with other variables in the dataset, and removing observations with missing values may lead to a biased sample that does not accurately represent the population. Secondly, some simple imputation (min, mean, or median) is a simple and fast method that replaces missing values with the minimum, mean, or median of the available values. The mean or median imputation assumes that missing values are missing completely at random (MCAR) and that the mean or median accurately represents the missing values. However, mean or median imputation can lead to biased estimates if the missing values are not MCAR, and it can also reduce the variability of the data and underestimate the standard errors for mixed-type

data. The min imputation assumes that missing values are missing not at random (MNAR), which means that the probability of missingness is only related to the unobserved missing values of the variable. It may sound reasonable that min imputation would work well if missingness occurs for certain proteins, which might not be quantified in specific conditions because they are below the detection limit in these specific samples. Unfortunately, it is not possible to test the MNAR until the missing values are known. Therefore, imputing the missing values with the minimum values will result in distorting the original distribution of the mixed-type variables. Lastly, the PMM method also assumes all missing values are MAR or MCAR and would fail if this assumption does not hold. The PMM method might be more appropriate than the regression method, which assumes a joint multivariate normal distribution if the normality assumption is violated. Marshall et al. concluded that in addition to producing less biased estimates, PMM produced better measures of model performance [11]. They also found that PMM might be the best approach when less than 50% of cases have missing data and are not MNAR [12].

Machine learning has widely been used to improve the accuracy of imputation techniques by building models that can predict the missing values based on the available data [13]. The basic idea behind this approach is to use the available data to train a model that can learn the underlying patterns in the data and use this knowledge to make predictions about the missing values. There are several machine learning algorithms that can be used for missing data imputation, including k-nearest neighbors, decision trees, regression methods such as classification and regression trees (CART) [14], and random forest, such as missForest [15]. The CART is an algorithm for the decision tree. Moreover, missForest is a non-parametric imputation method based on random forest. And it is applicable to various variable types and also robust to noisy data and multicollinearity due to the random forest's built-in feature selection [16].

On the one hand, there are several advantages of using machine learning techniques to handle missing values in data. First, they may reduce bias because the missing values may not be random and could be associated with certain groups or characteristics. A machine learning model can learn the patterns well from available datasets and make correct predictions for missingness and, therefore, provide better imputation results than traditional methods [17]. Secondly, some ML algorithms are robust to missing values. For example, the k-NN algorithm has the ability to exclude a column from the distance measure calculation when a value is missing. Similarly, naive Bayes can support missing values during the prediction process. Another algorithm that can be a suitable option for working with datasets that have null or missing values is random forest, which works well on non-linear and categorical data. Prediction of missing values depending on the nature (categorical or continuous) of the features having missing values or the models from data containing missing values can extend the performance for imputation. Lastly, they improve computational efficiency since some MLs are already embedded with high-performance computing (for example, supporting both CPUs and GPUs), which largely reduces computation time.

On the other hand, using machine learning to handle missing values also faces some challenges: For example, the machine learning-based imputation methods such as CART and missForest might produce inaccurate estimates of missing values, lead to overfitting, and cause model instability if they disregard or are inconsistent with the mechanisms of missingness for blood biomarkers in HABS–HD [8]. In order to overcome these challenges, we adopted multiple imputations to rely on machine learning models to predict missing values and incorporate uncertainty through different model characteristics.

Therefore, we proposed a three-step workflow embedded with a machine learning-based multiple imputation method (MLMI) to handle missing data in HABS–HD. Initially, we examined the missingness patterns in the HABS–HD dataset. Subsequently, we performed MLMI to fill in the missing values. Ensembling four machine learning models together and adapting them to multiple imputations with the averaging method provided a powerful machine learning technique that combines the predictions of multiple individual models to improve overall performance and generalization for missing data imputation. Finally, we

assessed and contrasted the performance of our imputation method with other prevalent imputation methods. Our results showed that the three-step workflow and the ML-based multiple imputation method performed well in handling missing values in HABS–HD.

## 2. Materials and Methods

### 2.1. Blood Collection and Processing

MCI and AD diagnoses were made according to the National Institute of Neurological and Communicative Disorders and Stroke and the Alzheimer's Disease and Related Disorders Association (NINCDS–ADRDA) criteria [18,19], which are known for their high reliability and validity [20]. These criteria involve impairment in eight cognitive domains and a decline in functional abilities, as assessed through neuropsychological testing. Participants who performed within expected ranges based on age and education level and did not meet the NINCDS–ADRDA criteria for cognitive impairment were considered normal controls.

Fasting blood samples were collected in accordance with the international guidelines for MCI and AD biomarker studies. The previously validated proteomic profile was then analyzed using electrochemiluminescence (ECL) as described in our published methods [21,22]. The measured biomarkers included fatty acid binding protein 3 (FABP3); beta 2 microglobulin (B2M); C-reactive protein (CRP); thrombopoietin (TPO); alpha 2 macroglobulin (A2M) eotaxin 3; tumor necrosis factor-alpha (TNFa); tenascin C (TNC); interleukin (IL)-5, IL-6, IL-7, IL-10, IL-18; I-309; factor VII (factor 7); soluble intercellular adhesion molecule 1 (sICAM1); circulating vascular cell adhesion molecule 1 (sVCAM1); and pancreatic polypeptide (PPY) as well as glucagon-like peptide 1 (GLP-1), insulin, homeostatic model assessment of insulin resistance (HOMA-IR), glucagon, and peptide YY (PYY). The measurement of plasma $A\beta40$, $A\beta42$, total tau (using a 3-plex plate), and NfL was performed with the Quanterix Simoa HD-1 platform. Based on our previous studies, we selected the proteins, validating their utility across platforms, tissues, and species [23,24].

### 2.2. A Three-Step Workflow to Handle Missing Data in HABS–HD

A three-step procedure was considered to handle missing data in HABS–HD (Table 1). The first step in handling missing data was to evaluate missing data, which includes exploring the missingness of variables, analyzing data distribution and correlation, and testing missing mechanisms. The second step was to perform imputation with ML-based multiple imputation (MLMI) and other imputation methods, for example, min, mean, predictive mean matching (PMM) [9,10] in mice, classification and regression trees (CART) [14] in mice, and missForest [15]. The last step was to evaluate the imputation. Three criteria were used for this purpose: (1) performance comparison among different imputation methods, (2) simulation evaluation, and (3) comparative analysis for data distribution and correlation before and after imputation. In the simulation evaluation, we initially acquired a complete dataset by implementing listwise deletion to handle missing values. Subsequently, we systematically introduced missing values into the samples, mirroring the original distribution of missingness. Finally, we evaluated the accuracy of the imputation process by comparing real values with imputed values using the root-mean-square error (RMSE).

**Table 1.** A three-step approach to deal with missing data in HABS–HD.

| Step | Action | Main Objective |
| --- | --- | --- |
| 1 | Missing data evaluation | missingness of variables analysis<br>data distribution and correlation analysis<br>missing mechanism testing |
| 2 | Imputation | min, mean, predictive mean matching (PMM) in mice, classification and regression trees (CART) in mice, missForest, and ML-based multiple imputation (MLMI) |

**Table 1.** *Cont.*

| Step | Action | Main Objective |
|:---:|:---:|:---|
| 3 | Imputation evaluation | performance comparison among different imputation methods<br>comparative analysis for data distribution and simulation evaluation<br>correlation before and after imputation |

### 2.3. Statistical Testing for the Missing Mechanism: MCAR Testing

One important consideration in selecting a missing data approach is determining the mechanism of missingness since different approaches make different assumptions. Following Enders (2010) [6], we defined the three general types of missing data mechanisms as follows:

$$\text{Missing completely at random (MCAR)} : p(M|Y) = p(M)$$
$$\text{Missing at random (MAR)} : p(M|Y) = p(M|Y_{obs})$$
$$\text{Missing not at random (MNAR)} : p(M|Y) = p(M|Y_{obs}, Y_{mis})$$

where data $Y$ contain completely observed variables $Y_{obs}$ and partly missing variables $Y_{mis}$, $Y = (y_{ij}) = (Y_{obs}, Y_{mis})$, and $M$ is a missing data indicator matrix, $M_{ij} = \begin{cases} 1 \text{ missing} \\ 0 \text{ observed} \end{cases}$. Briefly, if the probability of missing data on a variable is unrelated to any other measured variable and is also unrelated to the variable with missing values itself, then the missing data mechanism is considered MCAR; if the probability of missing data on a variable are related to some other measured variable in the model, but not to the value of the variable with missing values itself, then the missing data mechanism is assumed to be MAR; and if the missing values on a variable are related to the values of that variable itself, even after controlling for other variables, then the missing data mechanism may be MNAR.

Testing the MCAR mechanism can be performed using Little's MCAR test [25] or simple statistics testing by comparing the group with missing data to the group without missing data. However, it is not possible to distinguish between MNAR and MAR unless the missing values are known. The essence of testing for MCAR is to compare the group with missing data to the group without missing data. This can be accomplished using *t*-tests to compare for continuous measures and the chi-squared tests to compare for categorical measures. The smaller the *p*-value (<0.001), the stronger the rejection of the null hypothesis, indicating that the missing data are not MCAR (i.e., are either MAR or MNAR).

### 2.4. ML-Based Multiple Imputation (MLMI)

Based on the four machine learning models ((support vector machine (SVM), random forest (RF), extreme gradient boosting (XGB), and lasso and elastic-net regularized generalized linear model (GLMNET))), we developed a ML-based multiple imputation method (Figure 1, [26]). First, the four ML-based methods above were used for multiple imputation, and the results were pooled. Second, multiple generated estimates of missing values were averaged as the final imputation value. Lastly, the imputation quality and performance were compared to common imputation methods on our HABS–HD datasets. The comparison included (1) the effect on the performance of downstream machine learning tasks and (2) the effect on the distribution and correlation of variables before and after imputation. The commonly used imputation methods for the purpose of evaluation and comparison included listwise deletion, min/mean imputation, predictive mean matching (PMM) [9,10], classification and regression trees (CART) [14], and the single RF-based imputation method such as missForest [15].

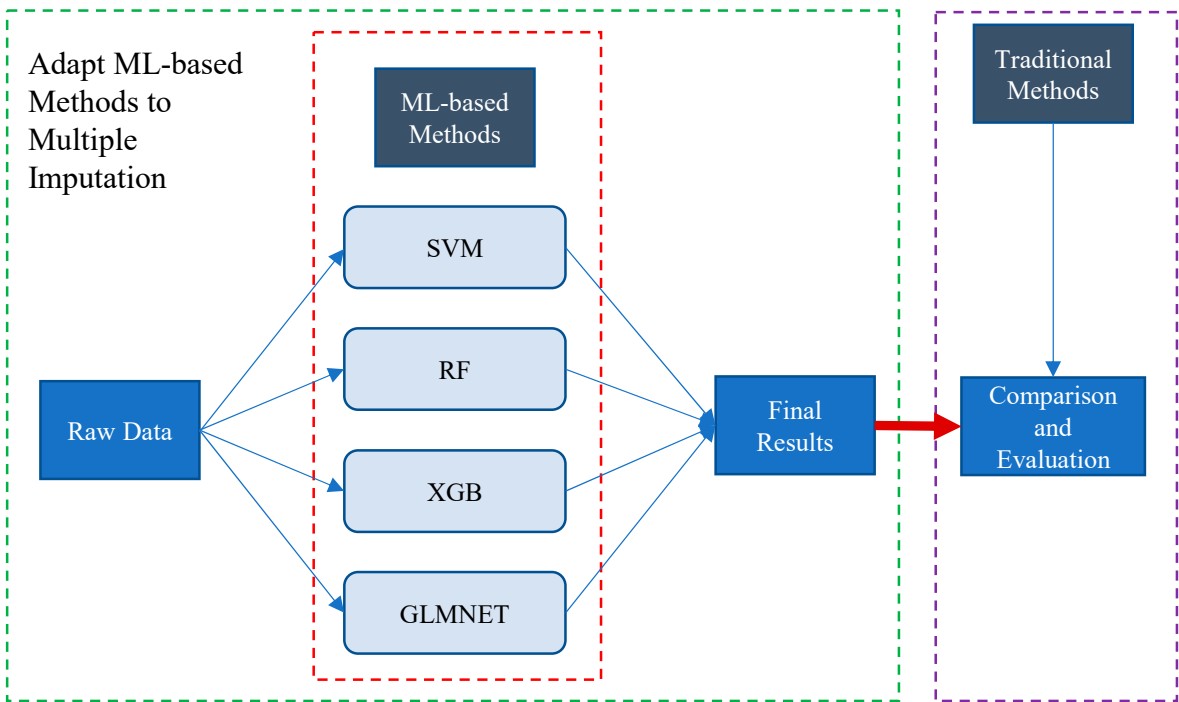

**Figure 1.** Flowchart for ML-based multiple imputation.

The e1071 package (v1.7-13) was used to perform support vector regression (SVR) for missing data imputation. SVR works on similar principles as support vector machine (SVM) classification. SVR is adapted from SVM when the dependent variable is numerical rather than categorical. A major benefit of using SVR is that it is a non-parametric technique. In contrast to simple linear regression, SVR does not rely on the distributions of the dependent and independent variables to yield its results. The randomForest package (V4.7.1) in R with default parameters was used for imputing missing data. The Xgboost package (V1.7.3) in R was used for regression imputation. A root-mean-squared error (RMSE) was used to evaluate the metric to build the model. To fit a linear regression via least squares, we also used the Gaussian family from the glmnet package (V4.1-6). λ in the logistic regression model was selected through lambda.min from cv.glmnet for the lasso tuning parameter λ when the cross-validated mean-squared error (MSE) was minimized.

*2.5. Performance Measurements*

The following eight performance measurements were involved in our evaluation and expected to be a more accurate estimate of the true unknown underlying mean performance of the model on the dataset: (1) sensitivity; (2) specificity; (3) precision; (4) accuracy; (5) negative predictive value, (6) positive predicted value at the base rate of 12%, (7) negative predicted value at the base rate of 12%, and (8) area under the curve.

**3. Results**

The HABS–HD data downloaded in March 2022 contained 1328 normal controls and 377 MCIs and ADs (116 ADs and 261 MCIs). Detailed demographic characteristics of the cohort are presented in Table 2. Detailed information about the HABS–HD dataset was described in the references [2,27]. The majority of the study cohort was female (61%), and just over half were older than 65 years of age (53.3%, *p*-value = $2.94 \times 10^{-12}$) and Hispanic (52%). The normal control group had a higher education level than the other two groups, MCI and AD (*p*-value = $1.30 \times 10^{-7}$). Significant differences were found in the distribution of age, gender, and Hispanic status between AD, MCI, and NC (*p*-value = 0.002 for age; *p*-value < 0.001 for gender; and *p*-value < 0.001 for Hispanic).

**Table 2.** Demographic characteristics of the cohort.

| Characteristic | Overall | AD | MCI | NC | *p*-Value [2] |
|---|---|---|---|---|---|
| N | 1705 [1] | 116 [1] | 261 [1] | 1328 [1] | |
| Age | 66.47 (8.75) | 69.62 (9.58) | 66.93 (9.24) | 66.10 (8.51) | 0.002 |
| Gender | | | | | <0.001 |
| F | 1034/1705 (61%) | 58/116 (50%) | 129/261 (49%) | 847/1328 (64%) | |
| M | 671/1705 (39%) | 58/116 (50%) | 132/261 (51%) | 481/1328 (36%) | |
| Hispanic | | | | | <0.001 |
| Hispanic | 890/1705 (52%) | 67/116 (58%) | 164/261 (63%) | 659/1328 (50%) | |
| Not Hispanic | 815/1705 (48%) | 49/116 (42%) | 97/261 (37%) | 669/1328 (50%) | |
| Education | 12.34 (4.82) | 10.62 (5.35) | 11.41 (4.86) | 12.68 (4.71) | <0.001 |
| CRP | 42,875,365.58 (67,985,315.03) | 31,996,905.71 (60,624,287.52) | 43,972,990.63 (70,617,506.74) | 43,606,283.11 (68,035,930.76) | 0.043 |
| (Missing) | 52 | 4 | 10 | 38 | |
| FABP3 | 5079.65 (2543.46) | 5314.49 (2937.28) | 5465.57 (2902.67) | 4984.40 (2423.31) | 0.033 |
| (Missing) | 49 | 4 | 10 | 35 | |
| IL_10 | 0.43 (0.65) | 0.38 (0.55) | 0.47 (0.71) | 0.42 (0.64) | 0.13 |
| (Missing) | 51 | 5 | 10 | 36 | |
| IL_6 | 1.90 (19.94) | 1.48 (2.04) | 1.53 (2.25) | 2.01 (22.53) | <0.001 |
| (Missing) | 51 | 5 | 10 | 36 | |
| Ab40 | 252.71 (68.40) | 260.73 (83.08) | 258.92 (73.22) | 250.81 (65.93) | 0.38 |
| (Missing) | 73 | 6 | 13 | 54 | |
| Ab42 | 12.07 (3.36) | 11.98 (4.00) | 12.46 (3.57) | 12.00 (3.25) | 0.063 |
| (Missing) | 74 | 6 | 13 | 55 | |
| Tau | 2.48 (1.09) | 2.72 (1.24) | 2.58 (1.65) | 2.44 (0.93) | 0.15 |
| (Missing) | 73 | 6 | 13 | 54 | |
| NFL | 19.35 (13.85) | 28.30 (22.12) | 21.40 (18.39) | 18.18 (11.37) | <0.001 |
| (Missing) | 80 | 7 | 14 | 59 | |
| PPY | 645.76 (483.14) | 769.98 (418.28) | 653.65 (355.85) | 633.74 (507.65) | 0.003 |
| (Missing) | 650 | 46 | 103 | 501 | |
| sICAM_1 | 2,433,486.97 (2,772,190.70) | 2,252,975.50 (3,169,549.76) | 2,419,840.53 (3,191,712.63) | 2,451,832.39 (2,646,899.66) | 0.19 |
| (Missing) | 55 | 4 | 11 | 40 | |
| sVCAM_1 | 3,804,308.06 (4,382,426.22) | 3,692,440.22 (5,141,564.85) | 3,585,681.60 (4,284,327.03) | 3,856,260.88 (4,331,636.05) | 0.97 |
| (Missing) | 55 | 4 | 12 | 39 | |
| TNF_alpha | 3.66 (15.36) | 3.23 (0.99) | 3.33 (1.30) | 3.76 (17.37) | 0.007 |
| (Missing) | 51 | 5 | 10 | 36 | |
| GLP_1 | 1.39 (2.29) | 1.72 (4.21) | 1.36 (1.81) | 1.37 (2.17) | 0.68 |
| (Missing) | 401 | 38 | 67 | 296 | |
| Glucagon | 65.02 (49.68) | 62.22 (40.38) | 67.82 (47.71) | 64.69 (50.69) | 0.47 |
| (Missing) | 448 | 43 | 69 | 336 | |
| PYY | 44.92 (39.20) | 45.62 (31.21) | 42.21 (31.45) | 45.38 (41.03) | 0.41 |
| (Missing) | 399 | 37 | 67 | 295 | |
| Insulin | 300.07 (298.36) | 240.21 (213.12) | 315.04 (354.89) | 301.78 (291.93) | 0.12 |
| (Missing) | 401 | 38 | 67 | 296 | |
| HOMA_IR | 2.47 (3.14) | 1.95 (1.84) | 2.70 (4.44) | 2.46 (2.91) | 0.29 |
| (Missing) | 409 | 41 | 68 | 300 | |

[1] Mean (SD); n/N (%); [2] Kruskal–Wallis rank sum test; Pearson's chi-squared test.

We considered a three-step method to handle missing data in HABS–HD (Table 1). In the first step, we evaluated the missingness of HABS–HD. There were seventeen blood marker variables chosen as predictors for the status of prevalent MCI and AD: CRP, FABP3, IL_10, IL_6, Ab40, Ab42, Tau, NFL, PPY, sICAM_1, sVCAM_1, TNF_alpha, GLP_1, glucagon, PYY, insulin, and HOMA_IR with age, gender, Hispanic (Hispanic or not), and Edu (years of education) added as covariates.

First, we analyzed the missingness of the 17 blood marker variables. Figure 2 and Table 2 show that 818 observations had no missing values. In PPY, 650 (38.12%) observations had missing values. Also, there were 448 (26.27%) missing values for glucagon. We removed

the two variables from the dataset first as they had more than 25% missing and could bias
the imputation. There is no universal cutoff value for removing a variable with missing
values. We chose 25% as the threshold based on the following three criteria: (1) features
with a percentage higher than 75% of missing data will be removed [3,4,28]; (2) if features
with a higher percentage (higher than 50% but lower than 75%) of missing data still are
important or have a strong relationship with the outcome of interest, then we will choose
not to drop; and (3) if the missing percentage exceeds 25% but is below 50%, we will remove
the variables if they are not valuable.

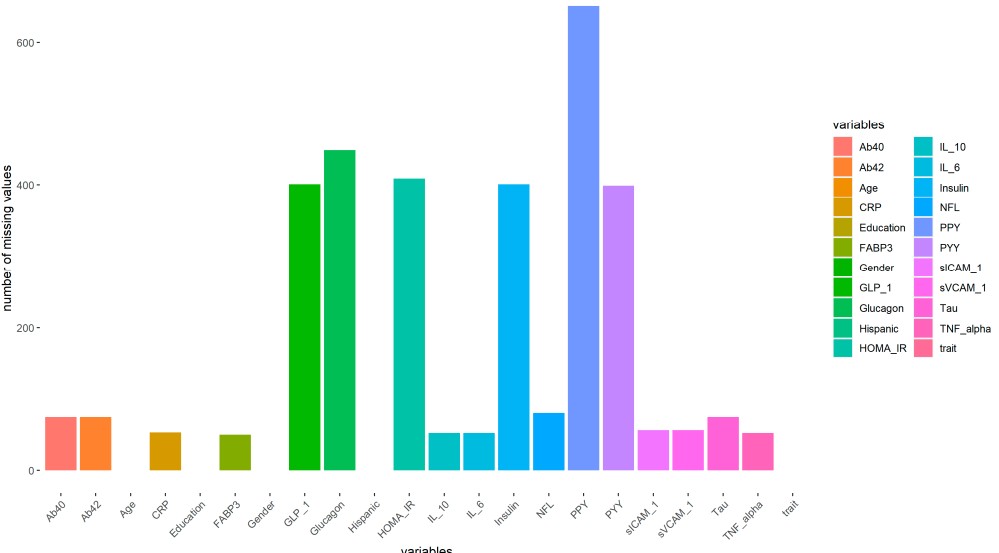

**Figure 2.** Number of missing values for variables in HABS–HD.

Secondly, we used the ggpairs() function from the GGally package (v 2.1.2) to explore
distributions and correlations by building and visualizing a scatterplot for the remaining
15 variables with correlation values (Figure 3). A statistically significant strong correlation
between insulin and HOMA_IR (corr = 0.96, *p*-value < $2.2 \times 10^{-16}$) was observed. This is
because HOMA_IR marks both the presence and extent of any insulin resistance and reveals
the dynamic between baseline (fasting) blood sugar and the responsive hormone insulin.
Moreover, values that were missing in insulin were observed also missing in HOMA_IR.
Therefore, we deleted the variable HOMA_IR, which contained a higher percentage of
missing because only one of the highly collinear variables was enough to use.

Lastly, we used the mcar_test() function in the naniar package (v1.0.0) to perform
Little's MCAR test [25] for the 14 continuous marker variables. The *p*-value for the test
close to 0 was significant, indicating that the missingness did not seem to be completely
random. We could not rule out MNAR in this situation since we could not test the missing
values themselves.

We also used a two-sample *t*-test and chi-squared test to compare the continuous
measures and the categorical measures, respectively, for the group with missing data
to the group without missing data. The results of the MCAR testing showed if the in-
dependent variables (in rows) related to the probability of missing data (in columns)
(Figure 4). On the GLP_1 variable, the patients that had observed values differed signifi-
cantly from patients with missing values on Hispanic (*p*-value = $5.60 \times 10^{-4}$), education
(*p*-value = $3.07 \times 10^{-5}$), and Ab40 (*p*-value = $5.90 \times 10^{-22}$). On the PYY variable, the
patients that had observed values differed significantly from patients with missing val-
ues on Hispanic (*p*-value = $3.49 \times 10^{-4}$), education (*p*-value = $3.67 \times 10^{-5}$), and Ab40
(*p*-value = $2.68 \times 10^{-21}$). On the Insulin variable, the patients that had observed values dif-
fered significantly from patients with missing values on Hispanic (*p*-value = $2.79 \times 10^{-4}$),
education (*p*-value = $2.38 \times 10^{-5}$), and Ab40 (*p*-value = $2.85 \times 10^{-18}$). The missingness in
FABP3 on PYY, the missingness in sICAM_1 on CRP, and the missingness in sVCAM_1 on

CRP were significantly different with *p*-values of $8.04 \times 10^{-5}$, $1.52 \times 10^{-4}$, and $5.19 \times 10^{-4}$, respectively. The smaller the *p*-value (<0.001), the stronger the rejection of the null hypothesis, indicating that the missing data were not MCAR (i.e., either MAR or MNAR).

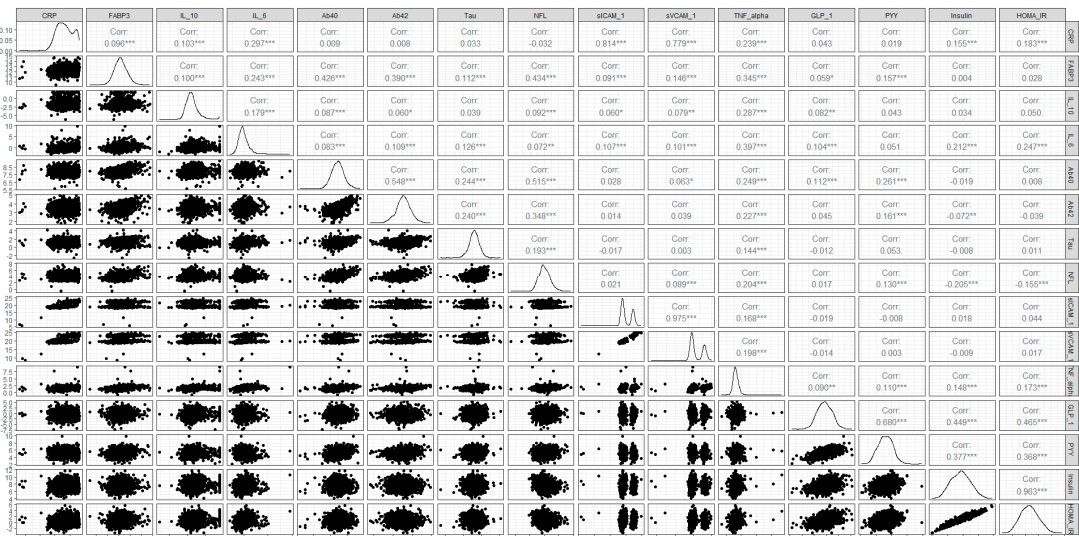

**Figure 3.** Distributions and correlations plot for variables. "***" if the correlation *p*-value is < 0.001, "**" if the correlation *p*-value is < 0.01, "*" if the correlation *p*-value is < 0.05, "." if the correlation *p*-value is < 0.10.

Based on Little's MCAR test [25] and the statistical MCAR testing from the first step, we concluded that the missingness in full HABS_HD data was mixed-type (eight were MCAR, and six were non-MCAR). Simply deleting all the missing values would not work in this situation. In the second step, we performed MLMI and other various imputation methods to estimate missing data. We used four R packages: e1071 package (v1.7-13), randomForest package (V4.7.1), Xgboost package (V1.7.3), and glmnet package (V4.1-6) to build the MLMI imputation model. The imputed values from the four models were averaged as the final estimation. In addition to simple statistics, min and mean, other packages we used were mice (v3.15.0) for predictive mean matching (PMM) [9,10] and classification and regression trees (CART) [14] and missForest [15] (v1.5) based on random forest.

In the third step, we evaluated these imputation methods via performance comparison and distribution and correlation change analysis. First, performance comparison among different imputation methods showed that of all compared performance results for AD vs. NC in the testing set of 10 times repeated 5-fold cross-validation, MLMI achieved the highest negative predictive value at base rate of 12% (93.82%) followed by PMM with 93.75%, cart with 93.43%, missForest with 93.37%, min with 93.24%, and mean with 93.22% (Table 3). Similarly, for MCI vs. NC in the testing set of 5-fold cross-validation, MLMI achieved the highest negative predictive value at a base rate of 12% (92.21%), followed by PMM with 91.57%, missForest with 91.48%, CART with 91.41%, min with 90.63%, and mean with 89.88% (Table 4). Secondly, in the simulation evaluation, MLMI achieved an RMSE of 1.178 with a standard deviation of 0.032, surpassing missForest (RMSE: 1.231, SD: 0.049), PMM (RMSE: 1.257, SD: 0.038), CART (RMSE: 1.313, SD: 0.038), mean (RMSE: 2.396, SD: 0.041), and min (RMSE: 3.533, SD: 0.036). Lastly, our results showed that there was no dramatic difference in data distribution and correlation before and after MLMI imputation (Figure 5, corr = 0.99, *p*-value < $2.2 \times 10^{-16}$). A dramatic change after missing data imputation would be viewed as a failure. This suggested that the MLMI method not only performed best among other imputation methods in machine learning prediction ability but also matched well with the original data patterns before imputation.

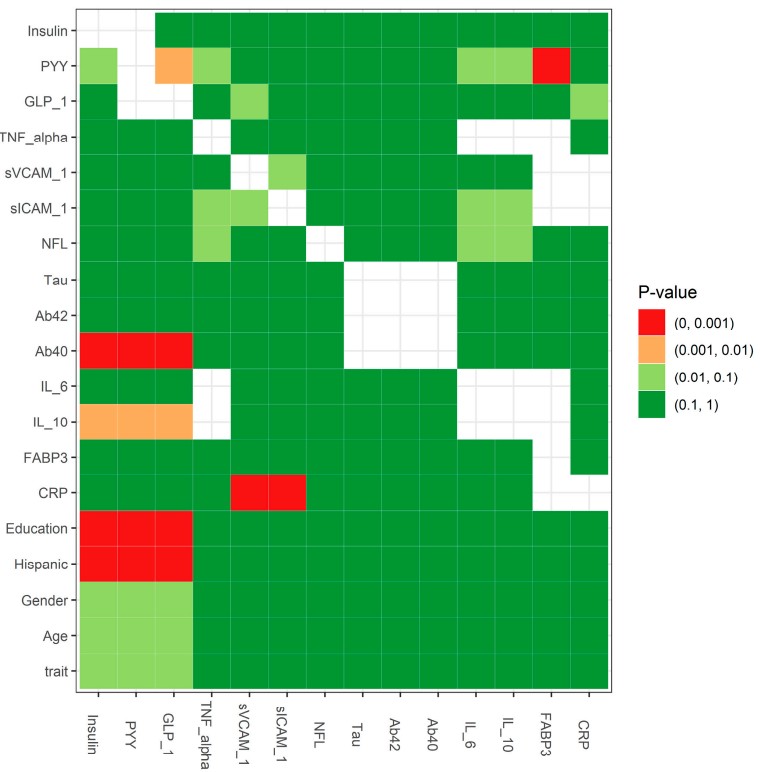

**Figure 4.** Statistical test for MCAR.

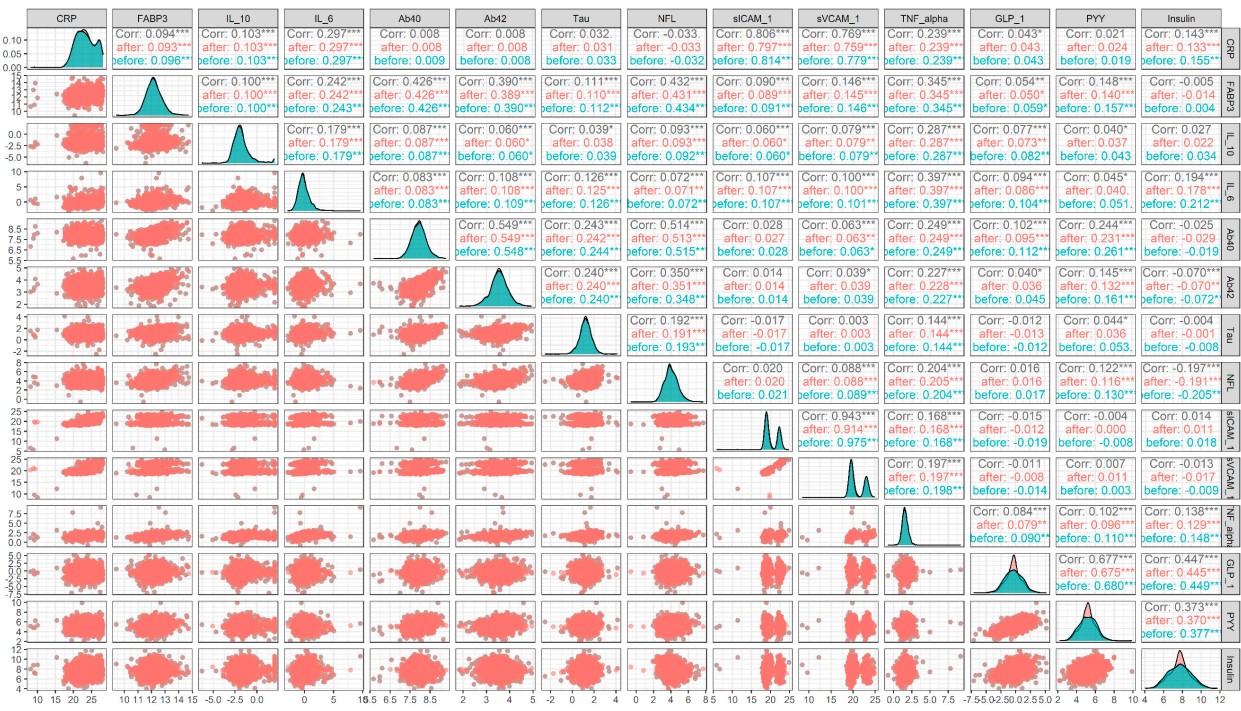

**Figure 5.** Distributions and correlations plot for variables before and after MLMI imputation. "***" if the correlation *p*-value is < 0.001, "**" if the correlation *p*-value is < 0.01, "*" if the correlation *p*-value is < 0.05, "." if the correlation *p*-value is < 0.10.

**Table 3.** Missing imputation methods comparison for predicting AD vs. NC.

| | Min | | Mean | | PMM | | CART | | missForest | | MLMI | |
|---|---|---|---|---|---|---|---|---|---|---|---|---|
| **Predicted** | **AD** | **NC** | **AD** | **NC** | **AD** | **NC** | **AD** | **NC** | **AD** | **NC** | **AD** | **NC** |
| AD | 14.436 | 79.304 | 14.296 | 76.884 | 15.052 | 77.668 | 14.54 | 76.124 | 14.628 | 79.644 | 15.028 | 74.92 |
| NC | 8.564 | 185.696 | 8.704 | 188.116 | 7.948 | 187.332 | 8.46 | 188.876 | 8.372 | 185.356 | 7.972 | 190.08 |
| Precision | 15.40% | | 15.68% | | 16.23% | | 16.04% | | 15.52% | | 16.71% | |
| Accuracy | 69.49% | | 70.28% | | 70.27% | | 70.63% | | 69.44% | | 71.22% | |
| Sensitivity | 62.77% | | 62.16% | | 65.44% | | 63.22% | | 63.60% | | 65.34% | |
| Specificity | 70.07% | | 70.99% | | 70.69% | | 71.27% | | 69.95% | | 71.73% | |
| NPV | 95.59% | | 95.58% | | 95.93% | | 95.71% | | 95.68% | | 95.97% | |
| AUC | 69.70% | | 70.55% | | 72.42% | | 72.30% | | 70.63% | | 72.63% | |
| PPV12 | 22.24% | | 22.61% | | 23.34% | | 23.08% | | 22.39% | | 23.96% | |
| NPV12 | 93.24% | | 93.22% | | 93.75% | | 93.43% | | 93.37% | | 93.82% | |

**Table 4.** Missing imputation methods comparison for predicting MCI vs. NC.

| | Min | | Mean | | PMM | | CART | | missForest | | MLMI | |
|---|---|---|---|---|---|---|---|---|---|---|---|---|
| **Predicted** | **MCI** | **NC** | **MCI** | **NC** | **MCI** | **NC** | **MCI** | **NC** | **MCI** | **NC** | **MCI** | **NC** |
| MCI | 30.104 | 117.812 | 26.36 | 106.804 | 34.412 | 132.32 | 33.008 | 124.628 | 34.176 | 131.968 | 35.596 | 130.02 |
| NC | 21.896 | 147.188 | 25.64 | 158.196 | 17.588 | 132.68 | 18.992 | 140.372 | 17.824 | 133.032 | 16.404 | 134.98 |
| Precision | 20.35% | | 19.80% | | 20.64% | | 20.94% | | 20.57% | | 21.49% | |
| Accuracy | 55.93% | | 58.22% | | 52.71% | | 54.69% | | 52.75% | | 53.81% | |
| Sensitivity | 57.89% | | 50.69% | | 66.18% | | 63.48% | | 65.72% | | 68.45% | |
| Specificity | 55.54% | | 59.70% | | 50.07% | | 52.97% | | 50.20% | | 50.94% | |
| NPV | 87.05% | | 86.05% | | 88.30% | | 88.08% | | 88.18% | | 89.16% | |
| AUC | 60.12% | | 58.55% | | 61.84% | | 61.38% | | 61.66% | | 63.39% | |
| PPV12 | 15.08% | | 14.64% | | 15.31% | | 15.54% | | 15.25% | | 15.98% | |
| NPV12 | 90.63% | | 89.88% | | 91.57% | | 91.41% | | 91.48% | | 92.21% | |

## 4. Discussion

### 4.1. Reasons for Missing Data in HABS–HD

There were several procedures in which missingness could occur for data collection in HABS–HD. For example, participants could fail to respond to questions, participants might provide biased input by responding to questions, or instruments might malfunction or not work ideally. Missing data in these surveys could introduce biased estimates and, therefore, incorrect conclusions. We addressed this by keeping surveys short and specific and conducting a training session for all participants, briefing them on all aspects of the study. Therefore, to increase the precision and clarity of our study, we directed our attention toward missing values that were specifically related to the blood biomarker collection process in HABS–HD. This allowed us to focus on missing data issues related to biomarkers.

### 4.2. Missing Data Mechanisms: MAR, MCAR, and MNAR

Among the three missing data mechanisms, MCAR, MAR, and MNAR, only MCAR can be tested. MAR vs. MNAR cannot be tested because information about MAR vs. MNAR is missing. HABS–HD data were tested to be mixed-type (eight variables were MCAR, and six variables were non-MCAR). Most simple statistic imputation (mean or median), multiple imputation, or regression imputation methods are based on MCAR assumption. Machine learning-based imputation methods such as missForest can be based on non-MCAR assumptions.

First, listwise deletion cannot be applied to the mixed-type data because it introduces bias in the results, and if using listwise deletion, we would lose approximately 26.3% of all samples from 1705 to 1257.

Secondly, neither min imputation nor mean imputation were ideally suited for HABS–HD (in AD vs. NC, NPV12 = 93.24% for min imputation and NPV12 = 93.22% for mean imputation; in MCI vs. NC, NPV12 = 90.63% for min imputation and NPV12 = 89.88% for mean imputation).

Thirdly, PMM, CART, and missForest imputation methods fit better than the min and mean imputation method (in AD vs. NC, NPV12 = 93.75% for PMM imputation, NPV12 = 93.43% for CART imputation, and NPV12 = 93.37% for missForest; in MCI vs. NC, NPV12 = 91.57% for PMM imputation, NPV12 = 91.41% for CART imputation, and NPV12 = 91.48% for missForest).

Lastly, in terms of mixed-type missing data imputation, MLMI outperformed PMM, CART, and missForest. The MLMI combined the advantages of the four ML models, SVM, RF, XGB, and GLMNET, which made it superior to regression-similar PMM imputation, CART imputation, and random forest-only missForest imputation. Moreover, the MLMI was a multiple imputation approach by replacing missing values with the mean of the four imputed values from the four imputation models, making it more robust than single imputation missForest when missing data were MNAR.

### 4.3. Limitations

The MLMI method ensembled four machine learning models together and adapted them to multiple imputations, therefore resulting in a more accurate imputation than the individual model. However, it has some limitations. First, the increase in performance of imputation depends on the complementarity of the individual model. If the individual models within an MLMI are not complementary to each other, meaning they do not offer different strengths or perspectives on the data, the MLMI may not perform significantly better than its individual components. To solve this problem, we established a three-step workflow to integrate MLMI with missing data evaluation and cross-validation to enhance complementarity. However, hyperparameter tuning and model selection, which could improve complementarity, are not performed in the current version of MLMI. The second limitation comes from the imputation evaluation. Neither the comparison of prediction performance nor the simulation evaluation are flawless. The former cannot evaluate the imputation of the blind testing set. The latter ignores the inherent uncertainty of missing values, and the measure based on similarity (RMSE) it uses between the true and imputed values may not separate valid from invalid imputation methods. The third limitation pertains to establishing an appropriate cutoff for changing data distribution and correlation before and after imputation, which often involves a trial-and-error approach.

### 4.4. Practical Considerations and Future Improvement

The MLMI method is straightforward, easy to implement, and not demanding in computation or resources compared to other common imputation methods. For example, for the HABS–HD dataset, it took about 34 s, running faster than missforest (57 s) and cart (48 s) but slower than pmm (6.5 s), min (0.33 s), and mean (0.31 s). MLMI also operated without a significant demand for extensive memory or processing power, the same as missforest, cart, pmm, min, and mean. A standard desktop computer/laptop with an i5 CPU and 8 GB of memory could handle it effectively.

Taking into account the independence of variables, future enhancements could involve implementing parallel programming for missing data imputation on a per-variable basis and adding hyperparameter tuning and model selection to improve complementarity in MLMI. Another improvement could be real-time imputation, which refers to the process of filling in missing or incomplete data points as soon as they occur, allowing data to be continuously updated and analyzed without delays. This method is invaluable in real-time data stream scenarios, such as online surveys or adaptive machine-learning applications. Real-time imputation algorithms function instantaneously as fresh data emerge, guaranteeing that datasets utilized for analysis or machine learning are consistently comprehensive and current, enabling the feasibility of adaptive machine learning. This

immediacy is particularly vital for AD monitoring mobile applications, ensuring prompt and precise responses and enabling well-informed decisions and actions based on the latest data.

## 5. Conclusions

In this paper, we presented a three-step workflow and a machine learning-based multiple imputation method for handling missing values in HABS–HD. Firstly, we conducted an in-depth investigation into the pattern, assumption, and potential causes and sources of the missing data present in the dataset. Secondly, we ensembled four machine learning models together and adapted them to multiple imputations with the averaging method. Finally, we evaluated and compared the performance of our imputation method with other commonly used imputation techniques. To do this, we assessed the prediction performance and the robustness of the distribution and correlation of each imputing method. We found that the three-step workflow embedded with the machine learning-based multiple imputation method was most appropriate for handling the mixed-type missingness for HABS–HD regarding prediction performance and match of distribution and correlation before and after imputation. It could help us overcome many challenges associated with data-driven biomarker discovery, disease diagnosis, and treatment outcome analysis of AD.

**Author Contributions:** Conceptualization, F.Z.; formal analysis, F.Z., M.P., S.E.O. and the Health and Aging Brain Study (HABS–HD) Study Team; investigation, F.Z., S.E.O. and the Health and Aging Brain Study (HABS–HD) Study Team; methodology, F.Z.; software, F.Z., M.P., S.E.O. and the Health and Aging Brain Study (HABS–HD) Study Team; validation, F.Z., M.P., R.F.P., S.E.O. and the Health and Aging Brain Study (HABS–HD) Study Team; writing—original draft, F.Z., M.P., L.J., J.H., R.F.P., S.E.O. and the Health and Aging Brain Study (HABS–HD) Study Team. All authors have read and agreed to the published version of the manuscript.

**Funding:** Research reported in this publication was supported by the National Institute on Aging of the National Institutes of Health under award numbers R01AG058537, R01AG054073, R01AG058533, 3R01AG058533-02S1, P41EB015922, and U19AG078109. The content is solely the responsibility of the authors and does not necessarily represent the official views of the National Institutes of Health.

**Institutional Review Board Statement:** The study was conducted in accordance with the Declaration of Helsinki and approved by the Institutional Review Board (or Ethics Committee) of UNTHSC (protocol code 2016-128 and 2020-125).

**Informed Consent Statement:** Informed consent was obtained from all subjects involved in the study. Written informed consent has been obtained from the patient(s) to publish this paper.

**Data Availability Statement:** The datasets for this study can be found in the datasets at https://apps.unthsc.edu/itr/ (accessed on 8 October 2023).

**Acknowledgments:** The authors acknowledge the Texas Advanced Computing Center (TACC) at the University of Texas at Austin in collaboration with the University of North Texas for providing the Lonestar6 computational and data analytics resources that have contributed to the research results reported within this paper.

**Conflicts of Interest:** S.E.O. has multiple pending and issued patents on blood biomarkers for detecting and precision medicine therapeutics in neurodegenerative diseases. He is a founding scientist and owns stock options in Cx Precision Medicine, Inc.

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
