# Peer review of "A Machine Learning-Based Multiple Imputation Method for the Health and Aging Brain Study–Health Disparities"

_informatics, doi:10.3390/informatics10040077_

Round 1

Reviewer 1 Report

In this paper, authors proposed a three-step workflow to handle missing data, and then applied the method in Alzheimer’s disease models. I think the paper is worth publishing. However, some points should be improved.

1. Missing values for some features could bring difficult for machine learning. Thus, topic in this paper is of importance. However, I want to know whether the data used in this paper has missing values. If there is, I think it is not enough to evaluate the method based on machine learning model. Authors should obtain a complete set of data without missing values, and then randomly delete some features from samples. Finally, authors should compare real values with filled Value.

2. The data obtained by this paper is from HSC. And in “Data Availability Statement”, authors still provided this link. Due to the fact that these data have been processed by the author, in order for more people to compare methods on the same data, the author should upload their processed data instead of the original data.

3. The original code should also be uploaded.

Author Response

In this paper, authors proposed a three-step workflow to handle missing data, and then applied the method in Alzheimer’s disease models. I think the paper is worth publishing. However, some points should be improved.

  1. Missing values for some features could bring difficult for machine learning. Thus, topic in this paper is of importance. However, I want to know whether the data used in this paper has missing values. If there is, I think it is not enough to evaluate the method based on machine learning model. Authors should obtain a complete set of data without missing values, and then randomly delete some features from samples. Finally, authors should compare real values with filled Value.

Answer: Thank you. We added simulation validation in the method section and result section.

“Three criteria were used for this purpose: 1) performance comparison among different imputation methods, 2) simulation evaluation, and 3) comparative analysis for data distribution and correlation before and after imputation. In the simulation evaluation, we initially acquired a complete dataset by implementing listwise deletion to handle missing values. Subsequently, we systematically introduced missing values into the samples, mirroring the original distribution of missingness. Finally, we evaluated the accuracy of the imputation process by comparing real values with imputed values using the Root Mean Square Error (RMSE).”

“Secondly, in the simulation evaluation, MLMI achieved an RMSE of 1.178 with a standard deviation of 0.032, surpassing missForest (RMSE: 1.231, SD: 0.049), PMM (RMSE: 1.257, SD: 0.038), CART (RMSE: 1.313, SD: 0.038), mean (RMSE: 2.396, SD: 0.041), and min (RMSE: 3.533, SD: 0.036).”

  1. The data obtained by this paper is from HSC. And in “Data Availability Statement”, authors still provided this link. Due to the fact that these data have been processed by the author, in order for more people to compare methods on the same data, the author should upload their processed data instead of the original data.

Answer: Thank you.

The processed data, in conjunction with the raw data, are housed within our HABS-HD repository and can be accessed for download via our website (see the link in the data availability). It is imperative that these datasets not be stored elsewhere, as doing so would violate the NIH-approved protocol.

  1. The original code should also be uploaded.

Answer: Thank you. We provided the R source codes for the ML-based Multiple Imputation (MLMI) (Figure 1) in GitHub and cited the URL as a reference. Simulated data was also provided in the GitHub URL for running the codes.

Reviewer 2 Report

The paper presents a three-step workflow and a Machine Learning-Based Multiple Imputation (MLMI) method for handling missing data in the Health and Aging Brain Study-Health Disparities (HABS-HD). The authors address the challenge of missing data in diverse community health studies, emphasizing the importance of considering different mechanisms of missingness. They propose the use of four ML-based models (SVM, RF, XGB, and GLMNET) for imputation and compare the performance of MLMI with other common imputation methods.

Strengths:

Relevance: The paper addresses a significant issue in health research by providing a robust method for handling missing data in the context of the HABS-HD study. Given the importance of accurate data analysis in healthcare, the research is highly relevant.

Comprehensive Approach: The authors systematically address different mechanisms of missingness (MCAR, MAR, MNAR) and propose a three-step workflow, which includes exploring missingness, developing ML-based imputation models, and evaluating their performance. This comprehensive approach enhances the paper's value.

Machine Learning Application: The use of Machine Learning techniques for imputation is a novel and promising approach, particularly given the increasing availability of large healthcare datasets. The inclusion of multiple ML models allows for a more robust evaluation of the proposed method's performance.

Comparative Analysis: The authors compare the MLMI method with other common imputation techniques, providing evidence of its superiority in terms of prediction performance and changes in distribution and correlation. This comparative analysis strengthens the paper's credibility.

Applicability: The paper emphasizes the potential applicability of the proposed method beyond Alzheimer's disease models, making it relevant to a broader range of health data analyses.

Areas for Improvement:

Clarity and Structure: While the paper is well-written, the structure could be further improved for clarity. It would be helpful to provide a concise overview of the three-step workflow at the beginning of the paper to help readers understand the research process at a glance.

Data Description: It would be beneficial to provide more details about the HABS-HD dataset, such as its size, characteristics, and types of missing data. A clear understanding of the dataset is essential for assessing the generalizability of the proposed method.

Discussion of Limitations: Although the paper focuses on the strengths of the proposed MLMI method, a more explicit discussion of its limitations and potential challenges would provide a more balanced view. For instance, discussing scenarios where ML-based methods might not be suitable would be valuable.

Practical Considerations: The paper could benefit from a section or discussion on practical considerations, such as computation time and resource requirements, when applying the MLMI method. This information is important for researchers considering its implementation.

In conclusion, the paper makes a significant contribution to the field of health research by proposing a Machine Learning-Based Multiple Imputation method for handling missing data in the HABS-HD study. The comprehensive approach, comparative analysis, and potential applicability of the method are strengths of the research. Addressing the suggested improvements would further enhance the clarity and completeness of the paper. Overall, this work has the potential to advance the field of healthcare data analysis and imputation methods.

None

Author Response

The paper presents a three-step workflow and a Machine Learning-Based Multiple Imputation (MLMI) method for handling missing data in the Health and Aging Brain Study-Health Disparities (HABS-HD). The authors address the challenge of missing data in diverse community health studies, emphasizing the importance of considering different mechanisms of missingness. They propose the use of four ML-based models (SVM, RF, XGB, and GLMNET) for imputation and compare the performance of MLMI with other common imputation methods.

Strengths:

Relevance: The paper addresses a significant issue in health research by providing a robust method for handling missing data in the context of the HABS-HD study. Given the importance of accurate data analysis in healthcare, the research is highly relevant.

Comprehensive Approach: The authors systematically address different mechanisms of missingness (MCAR, MAR, MNAR) and propose a three-step workflow, which includes exploring missingness, developing ML-based imputation models, and evaluating their performance. This comprehensive approach enhances the paper's value.

Answer: Thank you.

Machine Learning Application: The use of Machine Learning techniques for imputation is a novel and promising approach, particularly given the increasing availability of large healthcare datasets. The inclusion of multiple ML models allows for a more robust evaluation of the proposed method's performance.

Answer: Thank you.

Comparative Analysis: The authors compare the MLMI method with other common imputation techniques, providing evidence of its superiority in terms of prediction performance and changes in distribution and correlation. This comparative analysis strengthens the paper's credibility.

Answer: Thank you.

Applicability: The paper emphasizes the potential applicability of the proposed method beyond Alzheimer's disease models, making it relevant to a broader range of health data analyses.

Answer: Thank you.

Areas for Improvement:

Clarity and Structure: While the paper is well-written, the structure could be further improved for clarity. It would be helpful to provide a concise overview of the three-step workflow at the beginning of the paper to help readers understand the research process at a glance.

Answer: we added the following sentence in the abstract:

“Therefore, we proposed a three-step workflow to handle missing data in HABS-HD: 1) missing data evaluation, 2) imputation, and 3) imputation evaluation.”

Data Description: It would be beneficial to provide more details about the HABS-HD dataset, such as its size, characteristics, and types of missing data. A clear understanding of the dataset is essential for assessing the generalizability of the proposed method.

Answer: Thank you.

The Demographic characteristics of the cohort are listed in Table 1.

The missingness analysis is listed in Figure 2.

Missing types are listed in Figure 4.

We added the sentence in the results section.

“Detailed information about the HABS-HD dataset was described in the references [1] and [2].“

  1. O'Bryant, S.E.; Zhang, F.; Petersen, M.; Hall, J.R.; Johnson, L.A.; Yaffe, K.; Braskie, M.; Vig, R.; Toga, A.W.; Rissman, R.A.; et al. Proteomic Profiles of Neurodegeneration Among Mexican Americans and Non-Hispanic Whites in the HABS-HD Study. J Alzheimers Dis 2022, 86, 1243-1254, doi:10.3233/JAD-210543.
  2. O'Bryant, S.E.; Petersen, M.; Hall, J.R.; Large, S.; Johnson, L.A.; Team, H.-H.S. Plasma Biomarkers of Alzheimer's Disease Are Associated with Physical Functioning Outcomes Among Cognitively Normal Adults in the Multiethnic HABS-HD Cohort. J Gerontol A Biol Sci Med Sci 2023, 78, 9-15, doi:10.1093/gerona/glac169.

Discussion of Limitations: Although the paper focuses on the strengths of the proposed MLMI method, a more explicit discussion of its limitations and potential challenges would provide a more balanced view. For instance, discussing scenarios where ML-based methods might not be suitable would be valuable.

Answer: thank you. We added the following discussion about limitations in section 4.3.

“The MLMI method ensembled four machine learning models together and adapted them to multiple imputation, therefore resulting in more accurate imputation than the individual model. However, it has some limitations. First, the increase in performance of imputation depends on the complementarity of the individual model. If the individual models within an MLMI are not complementary to each other, meaning they don't offer different strengths or perspectives on the data, the MLMI may not perform significantly better than its individual components. To solve this problem, we established a three-step workflow to integrate MLMI with missing data evaluation and cross-validation to enhance complementarity. However, hyperparameter tuning and model selection which could improve complementarity are not performed in the current version of MLMI. The second limitation comes from the imputation evaluation. Neither the comparison of prediction performance nor the simulation evaluation is flawless. The former cannot evaluate the imputation of the blind testing set. The latter ignores the inherent uncertainty of missing values and the measure based on similarity (RMSE) it uses between the true and imputed values may not separate valid from invalid imputation methods. The third limitation pertains to the challenge of establishing an appropriate cutoff for changing data distribution and correlation before and after imputation, a task that often involves a trial-and-error approach.”

Practical Considerations: The paper could benefit from a section or discussion on practical considerations, such as computation time and resource requirements, when applying the MLMI method. This information is important for researchers considering its implementation.

Answer: Thank you. Please see the answer to the last comment.

In conclusion, the paper makes a significant contribution to the field of health research by proposing a Machine Learning-Based Multiple Imputation method for handling missing data in the HABS-HD study. The comprehensive approach, comparative analysis, and potential applicability of the method are strengths of the research. Addressing the suggested improvements would further enhance the clarity and completeness of the paper. Overall, this work has the potential to advance the field of healthcare data analysis and imputation methods.

Answer: Thank you.

We added a “Practical Considerations and Future Improvement” in the discussion section to discuss practical considerations and future improvement.

“The MLMI method is straightforward and easy to implement and not demanding in terms of computation or resources compared to other common imputation methods. For example, for the HABS-HD dataset, it took about 34 seconds, running faster than missforest(57 seconds), cart (48 seconds), but slower than pmm( 6.5 seconds), min (0.33 seconds), and mean(0.31 seconds). MLMI also operated without a significant demand for extensive memory or processing power as same as missforest, cart, pmm, min and mean. A standard desktop computer / laptop with an i5 CPU and 8GB of memory could handle it effectively.

Taking into account the independence of variables, future enhancements could involve implementing parallel programming for missing data imputation on a per-variable basis and adding hyperparameter tuning and model selection to improve complementarity in MLMI. Another improvement could be real-time imputation which refers to the process of filling in missing or incomplete data points as soon as they occur, allowing data to be continuously updated and analyzed without delays. This method is invaluable in scenarios involving real-time data streams, such as online surveys or adaptive machine learning applications. Real-time imputation algorithms function instantaneously as fresh data emerges, guaranteeing that datasets utilized for analysis or machine learning are consistently comprehensive and current, enabling the feasibility of adaptive machine learning. This immediacy is particularly vital for AD monitoring mobile applications, ensuring prompt and precise responses, enabling well-informed decisions and actions based on the latest data.”